# Scalable Early Childhood Reading Performance Prediction

**Zhongkai Shangguan[1]  Zanming Huang[1]  Eshed Ohn-Bar[1]**
**Ola Ozernov-Palchik[1,2]  Derek Kosty[3]  Michael Stoolmiller[1]  Hank Fien[1]**

[1]Boston University  [2]Massachusetts Institute of Technology  [3]Oregon Research Institute

## Abstract

Models for student reading performance can empower educators and institutions to proactively identify at-risk students, thereby enabling early and tailored instructional interventions. However, there are no suitable publicly available educational datasets for modeling and predicting future reading performance. In this work, we introduce the Enhanced Core Reading Instruction (**ECRI**) dataset, a novel large-scale longitudinal tabular dataset collected across 44 schools with 6,916 students and 172 teachers. We leverage the dataset to empirically evaluate the ability of state-of-the-art machine learning models to recognize early childhood educational patterns in multivariate and partial measurements. Specifically, we demonstrate a simple self-supervised strategy in which a Multi-Layer Perception (MLP) network is pre-trained over masked inputs to outperform several strong baselines while generalizing over diverse educational settings. To facilitate future developments in precise modeling and responsible use of models for individualized and early intervention strategies, our data and code are available at https://ecri-data.github.io/.

## 1 Introduction

Approximately 37% of fourth graders in the United States are not reading at grade level—a trend that persists into other grades [46]. As reading proficiency is closely correlated with academic, economic, and social outcomes [37], there is an urgent need to improve reading instruction practices. Currently, students are identified as needing additional educational support using a 'wait-to-fail' approach, i.e., waiting until a child has not made expected gains in reading before there is a re-evaluation of their instructional needs. A prediction model identifying which children would benefit from specific educational interventions could hold significant implications for achieving early and effective remediation. However, despite unprecedented interest in AI and education (e.g., the recently introduced AI Education Act [11]), there are no suitable publicly available datasets for training and evaluation of models for reading performance prediction and intervention success. As a result, predictive Machine Learning (ML) techniques have yet to be rigorously studied in early educational settings, such as for improving risk identification and proactively ensuring foundational reading abilities [12, 14, 36, 49, 51, 67].

Field datasets in early childhood education can be difficult to collect due to logistical factors related to working with young children, e.g., educators have limited time available for data collection. Students often transfer between schools and programs, making tracking difficult. Moreover, reliably assessing students with learning disabilities can present additional logistical issues, such as the need for specialized assessment tools and trained personnel to accurately evaluate and accommodate unique needs. These challenges have limited prior research to small-scale studies [15, 16, 25, 58] that did not release data nor can be leveraged to train high-capacity predictive ML models, i.e., for modeling complex relationships among multidimensional individual and instructional factors.

38th Conference on Neural Information Processing Systems (NeurIPS 2024) Track on Datasets and Benchmarks.

In this work, our goal is to develop a benchmark and framework for training accurate and precise ML models for predicting student reading performance. This includes predictability of responsiveness, i.e., leveraging data collected during well-validated first-grade reading interventions [6, 7, 24, 47, 55]. Our contributions are as follows: First, we introduce a longitudinal large-scale benchmark consisting of literacy assessment measurements from 6,916 first-grade students from 44 US schools. To ensure our findings are relevant to current intervention practices and generalize across settings, the data was collected as part of an IRB-approved study with an established multi-tiered intervention framework [6, 47]. Second, towards scalable modeling in realistic educational contexts, we perform a comprehensive comparative analysis among approaches for learning from tabular and missing data. Specifically, we demonstrate that predicting reading performance and outcomes for educational interventions involves a challenging modeling task, i.e., due to the inherent diversity in students with learning disabilities. To handle missing and diverse data, we demonstrate that a simple strategy for pre-training MLPs works particularly well in capturing complex interactions among various student, classroom, and school-level factors. As far as we are aware, we are the first to surface challenges in data-driven ML models that predict students' pre-to-post-intervention gains in large-scale and diverse early education generalization settings. The data and code will be made publicly available to facilitate future research in supporting early identification and individualized instructional support for students and teachers.

## 2 Related Work

Based on our survey of student reading assessment and modeling research, we identify a current gap in scalable modeling of reading skills in early education, which we discuss next.

**Reading Skill Assessment:** According to the National Assessment of Educational Progress (NAEP) [46] a significant portion of children in the US struggle with developing reading skills. Among all the states and jurisdictions that participated in a 2022 NAEP fourth-grade survey, the range of students performing below the NAEP basic level spanned 20-52%, with a national average of 39% for public school students. Therefore, it is critical to assess students' reading ability at an early stage, i.e., to inform the development of instructional strategies commensurate with identified areas of need to avoid students' reading deficiencies. Assessment of reading skills typically involves various word identification, spelling, sound-symbol knowledge, cloze tests, contextual reading, and text annotation tasks [5, 9, 29, 39, 62]. Within the context of our research, we focus on two fundamental literacy skills that young learners cultivate during their initial foray into the realm of reading and understanding language, i.e., ***word identification*** and ***word attack*** tasks [2, 6, 47]. More specifically, word identification entails the recognition of frequently occurring words through visual memorization, while word attack emphasizes a more challenging task of deciphering unfamiliar words by applying principles of phonics and phonetic knowledge.

**Enhanced Core Reading Instruction:** To facilitate predictive ML models that can ***predict student response to standard interventions***, we have incorporated in a subset of our data a common methodology for providing additional reading skill support in classrooms, referred to as the Enhanced Core Reading Instruction (ECRI). ECRI is a methodology that emphasizes prioritized content and structured teaching routines aimed at improving the quality of explicit instruction in Tier 1 settings, along with core-aligned small-group instruction in Tier 2 settings [6]. In Tier 1, ECRI enhancements prioritize instructional delivery of content related to beginning reading skills, including phonemic awareness, phonics, word reading, reading fluency, vocabulary, and comprehension. Students identified as at risk for reading difficulties receive the ECRI Tier 2 intervention. This intervention includes structured opportunities to preview and practice foundational skills such as phonemic awareness, phonics, word reading, and reading fluency, all aligned with Tier 1 instruction, delivered in a 30-minute small-group lesson. A key feature of ECRI is its dual focus: (a) enhanced core reading instruction in Tier 1, and (b) a supplemental Tier 2 small-group intervention for students at risk of reading difficulties. In our work, we collect intervention records and incorporate such information as additional context inputted to the model. We envision teachers employing our model to predict outcomes of interventions and thus inform tailored instructional strategies, e.g., providing additional reading support.

**Machine Learning for Education:** ML in education is currently receiving significant attention due to diverse potential use cases, such as personalized learning, early identification of at-risk students, automated grading, adaptive curriculum development, and more [11, 35, 66]. For instance, Kabakchieva [38] utilized machine learning techniques for content analytics to improve university student management. Similarly, Djambic et al. [20] proposed an adaptive ML system that utilizes reinforcement learning to create diverse learning materials tailored to individual learners' needs and conditions. There is a substantial amount of literature that focuses on forecasting students' future performance [1, 14, 20, 27, 36, 48, 60]. However, the application of ML methodologies has been primarily focused on higher education, with elementary and early education levels receiving less attention. According to a review by Xu et al. [61], only $1.69\%$ and $11.86\%$ of AI in education research between 2011 and 2021 focused on kindergarten and elementary school. Moreover, the exploration of ML in large-scale educational contexts with diverse data and student profiles remains limited. Consequently, the limitations of state-of-the-art deep learning-based approaches for modeling student reading skills in early stages remain mostly unexplored.

**Handling Missing Values in Tabular Data:** Educational datasets from real-world scenarios often suffer from a high incidence of missing values, particularly in large-scale data collection settings. This issue can be attributed to a variety of factors, such as unsuccessful data gathering (e.g., student absences during assessments or transfer to different schools) and errors in data entry [23, 50, 52]. To address learning from partial information in the data, traditional approaches generally impute the missing values, e.g., with a fixed constant and indicator. In recent years, the development of deep generative models has introduced approaches to generate missing entries based on available information [10, 54, 65, 68]. However, even with more advanced modeling techniques, directly imputing missing entries in the data can introduce noise and mask complex interrelationships between variables. This challenge becomes more pronounced with categorical data, such as gender or ethnicity, where imputed values need to be assigned to specific categories. In this work, we pursue an orthogonal direction by adopting a self-supervised model pre-training technique that enables robustness to the missing data by inferring missing information within an *embedding space* without explicitly filling in missing values. This technique allows for seamless integration of categorical and numerical data, as it avoids the hard-coded categorization of imputed values. In our analysis, we demonstrate that our pre-training with subsequent fine-tuning not only enhances data representation but also regularizes training to consistently improve model performance.

**Self-Supervised Learning in Tabular Data:** Given the novelty of the dataset, we aim to comprehensively analyze various models and training strategies for learning from tabular data. Due to the remarkable success of self-supervised learning in handling textual and visual data [22, 28, 41], researchers have recently started exploring self-supervised learning techniques for tabular data. VIME [64] applied self- and semi-supervised deep learning framework for tabular data, employing pretext tasks such as mask vector estimation and feature vector estimation to train an encoder that generates informative representations of raw input data. Similarly, SCARF [4] applies random feature corruption and introduces a self-supervised contrastive learning method that further demonstrates the potential of self-supervised learning techniques for tabular data. While we analyze such baselines within our reading performance prediction task, we also investigate their role when *learning from missing data*, which is under-discussed by prior literature. Specifically, we find that reported benefits of reconstruction loss-based VIME and contrastive and auxiliary losses in SCARF do not translate to our domain, i.e., to not outperform a simple MLP-based baseline that leverages self-supervised masked pre-training.

## 3 Dataset

Our goal is to surface limitations in scalable modeling of early childhood reading performances. In this section, we introduce the protocol and large-scale collection process used to source our novel dataset. The dataset will then be used to benchmark various ML methods in Sec. 5.

### 3.1 Enhanced Core Reading Instruction Dataset

Enhanced Core Reading Instruction, or ECRI, interventions have proven to be effective in supporting students who are at risk of reading difficulties [6, 24, 47, 55]. Specifically, first-grade students

who participated in ECRI were shown to exhibit marked improvement in key reading skills such as phonemic decoding, oral reading fluency, and word recognition.

**Data Collection:** Our data was collected as part of an IRB-approved study in 44 elementary schools. We collected student-level demographic data, including special education status, gender, and English proficiency status every year. Class size, school size, and the number of students at risk of reading failure are obtained from the school's enrollment records based on student participation in the fall screening assessment. Besides student and school attributes, our data includes student assessments, teacher surveys, and classroom observations. Assessors underwent comprehensive training, comprising three days of initial training on administration and scoring procedures prior to the fall assessment, supplemented by four additional days of training in the winter and spring. To ensure inter-rater reliability, assessment coordinators evaluated scoring consistency through shadow scoring and feedback on test administration. Teachers reported average daily instruction time and completed an online survey in the spring to assess their understanding of early reading instruction. Additionally, observations were conducted in treatment and control group classrooms during core reading instruction in November, February, and April, using Classroom Observations of Student Teacher Interactions [56], Ratings of Classroom Management and Instructional Support [21], and Quality of Explicit Instruction [47]. More details can be found in the supplementary material.

**Demographics:** Our study collected the dataset through an extensive study conducted over an entire academic year, involving 99 elementary schools across the United States. Throughout the study, $6,916$ first-grade students and 172 teachers were recruited and assigned into control and intervention groups through clustered randomization that nested students and teachers within schools. We de-identified all student and school names from our dataset to ensure privacy protection. Of the participants, $47.31\%$ underwent intervention, while the remaining $52.68\%$ did not. The gender distribution was $51.32\%$ male and $48.68\%$ female. The average percentage of Hispanic students was $20.40\%$ ($10.73\%$ intervention, $9.61\%$ control), the average percentage of Black American students was $3.39\%$ ($1.80\%$ intervention, $1.59\%$ control), and the average percentage of Asian American and Pacific Islander students was $6.96\%$ ($2.89\%$ intervention, $3.18\%$ control). Among our data samples, $48.61\%$ were eligible to receive Free and Reduced Price Lunch (FRL, $25.17\%$ intervention, $23.44\%$ control). We incorporate student and classroom-level variables into the model, including current reading level measures, age, gender, number of at-risk readers in the class, and a teacher knowledge score, as discussed next.

**Assessment:** The control group classes followed established core reading programs that aligned with the standard practices of their school districts, while *enhanced* core reading instructions were used in classes in intervention groups. Students underwent comprehensive reading assessments at the beginning and the end of the school year to evaluate various aspects of reading abilities. The assessments included Dynamic Indicators of Basic Early Literacy Skills (DIBELS) [30], Stanford Achievement Test [32], and Woodcock Reading Mastery Test (WRMT) [59]. We also included results from the Teacher Knowledge Survey (TKP) [45] and Ratings of Classroom Management and Instructional Support (RCMIS) [21] as measurements of teachers' understanding of reading instructions and teaching quality.

**Data Characteristics and Processing:** Missing data is prevalent in our dataset due to factors such as incomplete documentation or students being absent on the day of assessment. Out of 6,916 students and a total of eight reading assessment measurements, $30.48\%$ of the data is missing. While we retain the samples with missing data for pre-training, we exclude them for fine-tuning, evaluation, and analysis. Our analysis primarily focuses on students' progress on word identification and word attack tasks, given their strong correlation with overall student reading performance. The data is processed into a binary classification task to predict how well students responded to the intervention. Specifically, we use the average performance improvement of the control group over a school year as the reference, as most scores naturally increase at the end of the school year, and students with greater improvements are considered positive samples. Otherwise, we categorize them as negative samples. We note that this quantization occurs over the performance differences of all students, i.e., regardless of whether a student is from the intervention group or the control group, to indicate notable improvement in their performance. We also note that while this classification task is fundamental to determine whether additional support is required from the teachers in addition to the standard

intervention of more instructional time, our framework can be readily applied over more refined output quantization choices.

# 4 Method

In this section, we first outline the problem formulation, which involves predicting whether students will show improvement in reading ability. We then elaborate on the details of our simplified self-supervised pre-training methodology repurposed for handling missing data.

## 4.1 Problem Formulation

We consider the problem of learning to classify student progress from observed or partially observed variables $\mathbf{x} \in \mathbb{R}^d$ as learning a mapping function $f_{\boldsymbol{\Theta}}$, parameterized by weights $\boldsymbol{\Theta}$, to the probability $\mathbf{y} \in [0,1] \subseteq \mathbb{R}$ of likelihood of the student making sufficient progress, which in our case is defined as higher than the average in the control group for word identification and word attack tasks. We learn the model over all students, both in the control and intervention groups, and employ a set of $d = 16$ input variables, including the eight initial reading assessment measures (two

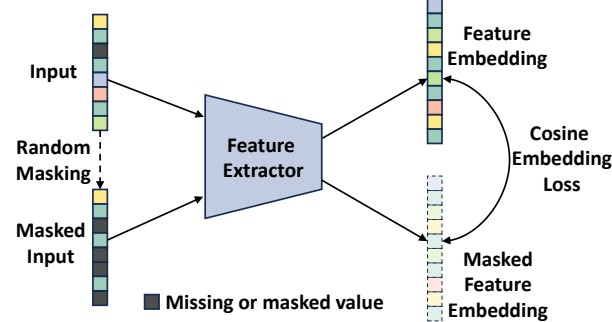

Figure 1: **Self-Supervised MLP Pre-Training.** We randomly mask parts of the input variables, i.e., as missing values, and train the model using a loss derived from both the original and masked input to a common feature extractor (we employ a cosine embedding loss to enforce similarity among the two embeddings).

from the Woodcock Reading Mastery test, two from the Nonsense Word Fluency test, one from the Oral Reading Fluency test, and three from a Stanford Achievement Test), as well as student and classroom-level variables, such as gender, intervention (i.e., additional reading instruction time) group status, age, number of at-risk readers in the class, and teacher rating based on an administered assessment. As noted in Sec. 3, missing data entries are common in our dataset, thus requiring special handling as discussed next.

## 4.2 MaskMLP Pre-Training

While missing data is common in real-world data collection settings, the component that is partially observed can still provide useful modeling cues. However, missing data entries are often imputed during training [44]. To better leverage information in all of the collected data, we analyze a pre-training technique, referred to as MaskMLP and depicted in Fig. 1. It is important to note that our method differs from the traditional approaches to handling missing data: we neither drop nor impute any missing values. Moreover, the approach simplifies self-supervised training methods in literature, such as VIME and SCARF. Instead, the pre-training approach enables the model to infer missing information within an *embedding space* without explicitly filling in missing values, which results in a smoother training process while maintaining the integrity of the original data structure.

For a given input vector $\mathbf{x}$, we first address missing features by marking them with an indicator value of $-1$. Subsequently, we randomly select a subset of observed features and mask them with $-1$ in order to generate the masked input $\mathbf{x}_{mask}$. The original input $\mathbf{x}$ and the masked input $\mathbf{x}_{mask}$ are both computed from the same student, enabling the model to learn relations among variables under partial observation.

The proposed training process draws inspiration from two aspects, the successful applications of pre-training strategies in language [18], vision [34], and tabular data domains [3, 4, 63, 64], as well as recent developments in deep learning demonstrating the alignment of multi-modal data in a shared high-dimensional embedding space [43, 53]. However, the aforementioned methods do not study the role of the masking mechanism and self-supervised pre-training process for benefiting from partially

observed data. We leverage a multi-layer perception (MLP) network that generates latent embeddings $\mathbf{E}_i \in \mathbb{R}^h$ and $\mathbf{E}_{mask} \in \mathbb{R}^h$ for the original input and masked input respectively, where $h$ denotes the hidden layer size. A cosine embedding loss is used for the pre-training objective,

$$\mathcal{L}_{pretrain}(\mathbf{E}_i, \mathbf{E}_{mask}) = 1 - \frac{\mathbf{E}_i \cdot \mathbf{E}_{mask}}{\|\mathbf{E}_i\| \cdot \|\mathbf{E}_{mask}\|} \qquad (1)$$

Following the pre-training phase, we augment the MLP architecture by attaching a classification head and fine-tuning the network for a binary cross-entropy loss [8].

## 5 Experiments

In this section, we leverage the introduced large-scale benchmark in order to uncover the limitations of ML-based models for the student reading performance prediction task, as well as validate the efficacy of self-supervised pre-training.

**Experimental Setup:** We use a group k-fold strategy [31] with grouping based on student and school ID, where $k = 5$. This ensures our partitioning maintains the integrity of student and school identifiers. We assign a value of $-1$ to missing variables during training, as none of our collected measurements are below zero. Thus, $-1$ effectively serves as an indicator [44], signaling the absence of a particular feature in a given data sample. Despite employing all data samples during pre-training, it is essential to highlight that no labels are used in the pre-training process. For MaskMLP, we randomly mask 25% of the observed variables as missing to compute the cosine embedding loss. In our large-scale benchmark, we provide a breakdown of both all students in the dataset and the intervention subgroup. This provides further insights into the effectiveness of the model over a group that receives additional reading instruction time. To produce a summarized analysis, we adopt accuracy and area under the precision-recall curve (AUC) [44] to evaluate model classification performances.

**Baselines:** Given the limited number of studies applying ML in our area, we also extensively benchmark diverse ML approaches that can be used for predicting early childhood education students' reading performance. Our analysis includes standard models such as logistic regression model [42], LightGBM [40], XGBoost [13], MLP [33], TabNet [3], VIME [64], SCARF [4], and MaskMLP. We also ablate several common strategies for handling missing data based on value replacement and indication [44]. In particular, following standard practices [44], we train our MLP using four different configurations: (1) measurements with zero-filling, (2) measurements with mean-filling, (3) incorporation of missing data indicators, and (4) utilization the self-supervised masked pre-training strategy (i.e., MaskMLP). We note that while TabNet [3] also employs a pre-training strategy, it is not effective for handling missing values. TabNet's pre-training requires an additional decoder, i.e., to reconstruct input variables where the pre-training loss is calculated using reconstructed and original values. Yet, as the latter may be partially missing in our settings, we instead leverage a loss defined over the *embedding space*, i.e., without requiring the complete original sample values.

### 5.1 Results

**Model Performances:** In our study, we utilize both demographic and assessment-based features to predict the specified target in word identification or word attack tasks, as described in Sec. 3.1. As illustrated in Table 1, MLP outperforms multiple baselines, including logistic regression, XGBoost, and LightGBM, as it can model complex relationships among the variables in the data without doing any feature engineering. In our experiments, we found a three-layer MLP model with a hidden size equal to 64 to perform best (complete ablations are provided in our supplementary document). In addition, the simplified pre-training is shown to consistently outperform the strong baselines of TabNet, VIME, and SCARF when learning from our noisy and partial data. Specifically, we find the additional optimization and auxiliary tasks, e.g., in the input space, to hinder leveraging missing information.

Furthermore, we analyze our method's impact on students who underwent an intervention of additional instructional time [6, 47] through targeted evaluation in Table 1. The analysis reveals that all approaches show improved performance in the intervention subset by 10% in both word identification and word attack tasks, suggesting the interventions create a more distinct feature space that positively

Table 1: **Results on the Large-Scale ECRI Dataset.** We report model performance for the two main reading skill prediction tasks over different generalization settings. School-split enforces disjoint schools in training and testing, while student-split allows students within the same school to be in the training and testing split. Model performances are shown for the entire set of test students and over the intervention subset (received additional instruction time). The best results are shown in **bold**.

| Method | Word Identification | | | | Word Attack | | | |
|---|---|---|---|---|---|---|---|---|
| | *All Students* | | *Intervention Group* | | *All Students* | | *Intervention Group* | |
| | Accuracy | AUC | Accuracy | AUC | Accuracy | AUC | Accuracy | AUC |
| *School Split:* | | | | | | | | |
| Logistic Regression [42] | 0.6119 | 0.6121 | 0.6214 | 0.5850 | 0.5923 | 0.5591 | 0.6311 | 0.5130 |
| XGBoost [13] | 0.6503 | 0.6515 | 0.6504 | 0.6536 | 0.6090 | 0.5817 | 0.6376 | 0.5628 |
| LightGBM [40] | 0.6651 | 0.6653 | 0.7086 | 0.6903 | 0.6223 | 0.5593 | 0.6958 | 0.5472 |
| TabNet [3] | 0.6385 | 0.6311 | 0.6925 | 0.6773 | 0.6328 | 0.5797 | 0.7120 | 0.5862 |
| GRAPE [65] | 0.6205 | 0.6238 | 0.6359 | 0.6281 | 0.6002 | 0.5426 | 0.6510 | 0.5349 |
| TabNet (Pretrain) [3] | 0.6458 | 0.6468 | 0.6926 | 0.6716 | 0.6208 | 0.5821 | 0.7186 | 0.5981 |
| VIME [64] | 0.6651 | 0.6585 | 0.7125 | 0.7012 | 0.6698 | 0.6322 | 0.7320 | 0.6281 |
| SCARF [4] | 0.6656 | 0.6586 | 0.7325 | 0.7222 | 0.6608 | 0.6258 | 0.7418 | 0.6602 |
| MLP (Zeros) | 0.6725 | 0.6670 | 0.7410 | 0.7319 | 0.6535 | 0.6303 | 0.7346 | 0.6369 |
| MLP (Mean) | 0.6695 | 0.6640 | 0.7281 | 0.7152 | 0.6371 | 0.5849 | 0.7476 | 0.6236 |
| MLP (Indicator) | 0.6710 | 0.6607 | 0.7539 | 0.7394 | **0.6698** | 0.6321 | 0.7412 | 0.6403 |
| **MaskMLP** | **0.6726** | **0.6693** | **0.7704** | **0.7633** | 0.6697 | **0.6545** | **0.7704** | **0.6869** |
| *Student Split:* | | | | | | | | |
| Logistic Regression [42] | 0.6207 | 0.6201 | 0.6474 | 0.6394 | 0.6281 | 0.5666 | 0.6796 | 0.5503 |
| XGBoost [13] | 0.6607 | 0.6603 | 0.6699 | 0.6659 | 0.6119 | 0.5743 | 0.6731 | 0.5849 |
| LightGBM [40] | 0.6711 | 0.6702 | 0.7121 | 0.6986 | 0.6385 | 0.5763 | 0.7022 | 0.5749 |
| TabNet [3] | 0.6519 | 0.6496 | 0.6989 | 0.6956 | 0.6356 | 0.5737 | 0.7184 | 0.5850 |
| GRAPE [65] | 0.6468 | 0.6402 | 0.6625 | 0.6570 | 0.6329 | 0.5976 | 0.6881 | 0.5972 |
| TabNet (Pretrain) [3] | 0.6652 | 0.6598 | 0.6892 | 0.6777 | 0.6563 | 0.6148 | 0.7021 | 0.6161 |
| VIME [64] | 0.6785 | 0.6763 | 0.7250 | 0.7193 | 0.6564 | 0.6121 | 0.7631 | 0.6930 |
| SCARF [4] | 0.6563 | 0.6549 | 0.7142 | 0.7112 | 0.6652 | 0.6372 | 0.7656 | 0.6944 |
| MLP (Zeros) | 0.6756 | 0.6745 | 0.7120 | 0.7002 | **0.6756** | 0.6368 | 0.7638 | 0.6956 |
| MLP (Mean) | 0.6652 | 0.6627 | 0.7186 | 0.7171 | 0.6563 | 0.6214 | 0.7572 | 0.6799 |
| MLP (Indicator) | 0.6889 | 0.6873 | 0.7411 | 0.7198 | 0.6637 | 0.6202 | 0.7670 | 0.6851 |
| **MaskMLP** | **0.6919** | **0.6899** | **0.7766** | **0.7537** | 0.6741 | **0.6485** | **0.7865** | **0.7333** |

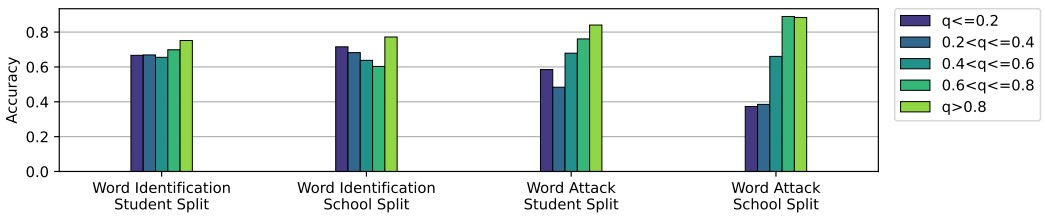

Figure 2: **Breakdown Results Across Quantile Groups on ECRI.** A five-class model accuracy breakdown with classes defined over five quantiles of student performance by improvement amount from the first assessment, from large regression, slight regression, no change, slight improvement, and large improvement.

affects students' reading skills (this underlines the significant role of such interventions). Our proposed MaskMLP method consistently achieves the best performance on the intervention subset, with larger accuracy and AUC gains over the baselines compared to results over the entire student set. We also conduct statistical tests to identify whether our model outperforms the other models by comparing MaskMLP against MLP and VIME over five different folds using a paired samples t-test [19, 17]. For the MLP baseline, the t-statistic was 3.876, and the p-value was 0.0090, suggesting that MaskMLP's improvements are statistically significant. For VIME [64], the t-statistic was 2.1023, and the p-value was 0.0517, indicating a trend toward significance. When comparing with SCARF [4], the t-statistic in word identification was 2.0954, and the p-value was 0.0521.

**Model Performance Across Student Progress:** We classify students into five distinct categories based on their progression quantiles: regression, slight regression, no change, slight improvement, and large progress. We then analyze classification performance separately for each of the five categories, with results shown in Fig. 2. We find our model to perform better among students who have made greater progress than those who have regressed, especially for the word attack task. In contrast, the

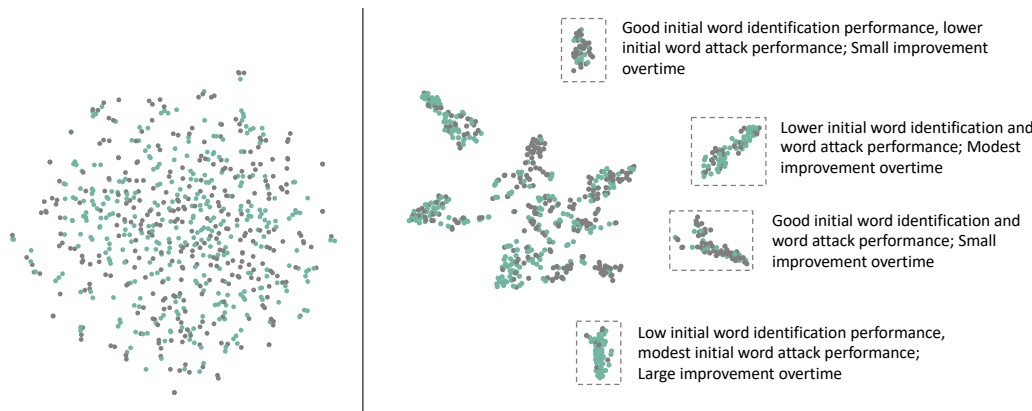

Good initial word identification performance, lower initial word attack performance; Small improvement overtime

Lower initial word identification and word attack performance; Modest improvement overtime

Good initial word identification and word attack performance; Small improvement overtime

Low initial word identification performance, modest initial word attack performance; Large improvement overtime

Figure 3: **Visualization of t-SNE-based Embedding and Student Profile Analysis.** The visualization uses embeddings derived from the MLP *(left)* and MaskMLP *(right)* models for the word identification task, with negative samples shaded in gray and positive samples shaded in green. The pre-training step in MaskMLP results in an embedding with greater separation among student profiles.

model exhibits lower success rates for student skill improvement prediction over the students with lower performance gains. This finding highlights a current model limitation that requires further study in the future, e.g., the need to carefully handle data imbalance among various potential student groupings and students.

**Feature Importance:** We systematically exclude each feature individually from ECRI and assess feature importance based on its impact on final model classification performance. We calculate the decrease in classification accuracy after excluding each input feature, with a larger decrease in accuracy indicating greater importance to the model. For classification on word identification task (Fig. 4), the word identification performance at the beginning of the school year is most influential, followed by other literacy evaluations including the SAT, DIBELS Oral Reading Fluency (ORF) and Nonsense Word Fluency (NWF) measures. To a lesser degree, classroom features such as teacher knowledge (TKP Score), classroom management behaviors (RCMIS Score), and fidelity to the implementation of ECRI curriculum (Fidelity) are also important contributors to dif-

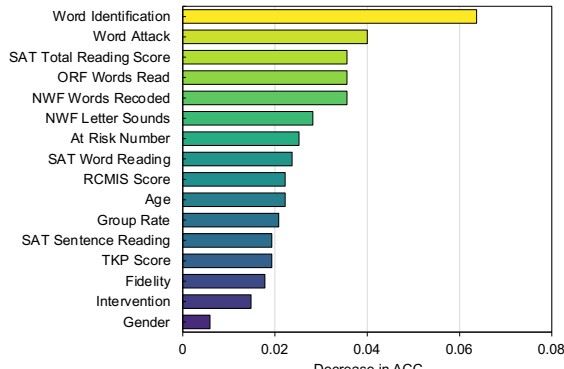

Figure 4: **Feature Importance Analysis.** We show feature importance by plotting the decrease in model accuracy on word identification classification after dropping each input variable, one at a time. All literacy assessment measures used for input are obtained at the start of the school year.

ferentiating student response. Additional ablations and results can be found in the supplementary.

**Impact of Pre-Training on Learned Embedding Space:** To better understand the role of our proposed pre-training method in the latent high-dimensional embedding space, we extract feature embeddings of all students from ECRI using MLP and MaskMLP and leverage a t-SNE [57] to project and visualize the high-dimensional embeddings in 2D space, as shown in Fig. 3. Notably, the feature embeddings extracted by MaskMLP exhibit separable and well-defined clusters, each representing a distinct student profile. MaskMLP is found to group similar features in embedding space without the need for explicit supervision, thereby enhancing downstream prediction performance.

**Bias Characterization:** To characterize predictive bias in our context, we provided a detailed breakdown of three specific demographic groups: gender, at-risk students, and socio-economic

Table 2: **Breakdown Based on Gender, At-Risk, and Socio-Economic Status.** We report a detailed breakdown of three specific demographic groups: gender, at-risk students, and socio-economic status of our MaskMLP model.

| Subset | Word Identification | | | | Word Attack | | | |
|---|---|---|---|---|---|---|---|---|
| | All Students | | Intervention Group | | All Students | | Intervention Group | |
| | Accuracy | AUC | Accuracy | AUC | Accuracy | AUC | Accuracy | AUC |
| *School Split:* | | | | | | | | |
| Overall | 0.6726 | 0.6693 | 0.7704 | 0.7633 | 0.6697 | 0.6545 | 0.7704 | 0.6869 |
| At-Risk | 0.6692 | 0.6672 | 0.7195 | 0.7140 | 0.6436 | 0.6170 | 0.7326 | 0.6429 |
| Male | 0.6753 | 0.6753 | 0.7784 | 0.7698 | 0.6736 | 0.6572 | 0.7585 | 0.6896 |
| Female | 0.6636 | 0.6606 | 0.7620 | 0.7521 | 0.6422 | 0.6486 | 0.7765 | 0.6754 |
| High FRL | 0.6200 | 0.5770 | 0.7500 | 0.7624 | 0.6281 | 0.6178 | 0.6042 | 0.5556 |
| Low FRL | 0.6874 | 0.6979 | 0.5873 | 0.6129 | 0.5574 | 0.5297 | 0.6785 | 0.6066 |
| *Student Split:* | | | | | | | | |
| Overall | 0.6919 | 0.6899 | 0.7766 | 0.7537 | 0.6741 | 0.6485 | 0.7865 | 0.7333 |
| At-Risk | 0.6691 | 0.6693 | 0.7096 | 0.7043 | 0.6707 | 0.6434 | 0.7558 | 0.6731 |
| Male | 0.6969 | 0.6968 | 0.7864 | 0.7666 | 0.6782 | 0.6496 | 0.7725 | 0.7202 |
| Female | 0.6728 | 0.6755 | 0.7620 | 0.7425 | 0.6706 | 0.6349 | 0.7886 | 0.7426 |
| High FRL | 0.6217 | 0.5717 | 0.7500 | 0.7579 | 0.6656 | 0.6178 | 0.6301 | 0.6048 |
| Low FRL | 0.6736 | 0.6417 | 0.7143 | 0.6667 | 0.5575 | 0.5298 | 0.7358 | 0.6128 |

status, as shown in Table 2. We found that despite a relatively balanced gender distribution in our dataset, some performance differences were observed between different groups. Additionally, our model demonstrates lower performance for at-risk students. This outcome is potentially due to the heterogeneity within the at-risk group, where multiple overlapping challenges (e.g., language barriers, learning disabilities, and socio-emotional issues) confound accurate prediction, highlighting the need for more comprehensive data and context-aware model design to enhance support for this population.

We further sought racial-ethnic and free/reduced-price lunch (FRL) data from participating schools and districts to investigate the relationship between socio-economic status (SES) and model performance. Due to confidentiality concerns, we opted to use and publish race-ethnicity and FRL data only at the school level. Our analysis reveals a consistent trend of higher model predictive performance in schools with higher SES (i.e., fewer FRL students). Notably, there were substantial SES and race/ethnicity differences between student groups (e.g., 8% versus 35% Hispanic, and 2% versus 7% African American students). Although there may be some external factors, such as students of higher socioeconomic status often benefit from additional resources beyond school instruction, such as private tutoring, we found that model performance on students from high socioeconomic status students was higher. Additionally, we observed that the effects of teacher knowledge and experience on student achievement were reduced in classrooms with the intervention. In contrast, the effects of teacher differences were more pronounced in classrooms without the intervention, which used a variety of instructional approaches common in U.S. schools. These findings suggest that the consistent use of evidence-based strategies can help reduce disparities in educational delivery and promote more equitable outcomes.

## 6   Limitations

Our models can predict the outcomes of instructional interventions, facilitating timely and appropriate actions, such as additional instructional support from teachers. However, model generalizability and robustness—such as performance across different schools and student profiles—remain limited to certain student groups. While our dataset and methods have the potential to improve educational outcomes, irresponsible use could negatively impact students' educational experiences and development. Given the inherent data and model biases, AI techniques may fail for atypical profiles and could perpetuate biases if not applied cautiously. By releasing publicly available data and open-source models, we aim to increase prediction transparency to help mitigate potential harms. Our models should be accompanied by educational materials, and we provide users with guidelines regarding common failure modes and potential risks, especially for rare student profiles from diverse educational and personal backgrounds.

We recognize the critical ethical and societal implications of using machine learning in education, especially regarding bias mitigation to minimize risks. We envision ML predictions as tools to assist teachers in identifying at-risk students for timely intervention. However, these benefits may not

be realized if the underlying datasets contain biases. Even with diverse and representative data, model failure or misuse can potentially harm students already at risk, such as through mislabeling or misidentification. We hope this dataset will contribute to a better understanding of bias issues at the intersection of machine learning and education and serve as a step toward addressing them.

## 7 Conclusion and Future Work

In our study, we explore challenges and opportunities in applying machine learning models for predicting early childhood reading performance. We develop a benchmark using multiple ML models on a comprehensive longitudinal dataset of literacy assessments and an evidence-based intervention for at-risk early learners. Despite the large proportion of missing data, we develop a self-supervised pre-training strategy, referred to as MaskMLP, which was shown to better handle missing values and provide accuracy gains compared to multiple state-of-the-art approaches. Our findings and insights surface several implications for advancing preventive approaches in education and improving the delivery of tailored support to address diverse needs across various educational settings, specifically for raising awareness of the current limitations of ML models over various evaluation settings and student types. We envision our models responsibly used to augment teacher decision-making processes, allowing them to provide timely support to enhance student reading and learning outcomes.

In future work, we plan to apply our study to educational environments in situ, which would allow us to investigate prediction uncertainty and information about model failure cases to better provide educators with effective information to improve timely support and facilitate responsible use given widespread bias issues in data-driven systems, particularly for rare and outlying data samples [26].

## Acknowledgments

The authors disclosed receipt of the following financial support for the research, authorship, and/or publication of this article: The research reported here was supported by the Institute of Education Sciences, U.S. Department of Education, through Grant R324A090104 to the University of Oregon, and a Red Hat Collaboratory research award. The opinions expressed are those of the authors and do not represent the views of the Institute of Education Sciences or the U.S. Department of Education.

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
