# OpenReview forum: "Scalable Early Childhood Reading Performance Prediction"
_NeurIPS.cc/2024/Datasets_and_Benchmarks_Track — NeurIPS 2024 Track Datasets and Benchmarks Poster_

### Official Review · Reviewer_pkea · 2024-07-15
**R1**

**Rating:** 7
**Confidence:** 4

**Review:**

Quality: The quality of the research is good, containing a large dataset and a comprehensive assessment of machine learning models. The methodology and the results are clearly presented.
Clarity: The paper is well written and structured, making it easy to follow the research objectives, methods and results.
Originality: This work is original as it introduces a novel data set and applies self-supervised learning techniques to predict early childhood reading performance.
Significance: The research is significant as it addresses a critical need in early childhood education by providing tools for early identification of at-risk students and enabling timely interventions.

Pros:
-	Introduction of a unique, large-scale data set.
-	Application of advanced machine learning techniques.
-	Potential for significant impact on early childhood education practice.
Cons:
-	Limited discussion of the potential negative impact of using machine learning in education.

**Strengths:**

The key strength of the work lies in its novel dataset and comprehensive approach to evaluating machine learning models for predicting early reading development. The importance of this work is highlighted by its potential to transform educational practice by enabling early intervention.

**Additional Feedback:**

1.	How do you plan to address potential bias in the data set and models?
2.	Can you provide more details on the consent process for data collection, in particular to protect the privacy of students and parents?
3.	What plans do you have for maintaining and updating the dataset in the future?
4.	Have you considered the potential long-term impact of predictive modeling on students' educational development and overall well-being?

**Clarity:**

The paper is well written, has a clear structure and a logical organization. The objectives, methods and results are presented in a concise way so that they are understandable for readers.

**Correctness:**

The claims in the submission seem to be correct based on the data and analysis provided. The accuracy of the various aspects of the submission is assessed below:

Structure of the dataset:
The ECRI dataset was constructed in a thoroughly using appropriate data collection and processing methods. The description of the dataset details the demographic characteristics, the variables collected and the longitudinal character of the data. The large sample size (6,916 students and 172 teachers at 44 schools) and the focus on first grade students provide a solid foundation for predictive modeling.

Experimental design and evaluation methods:
The experimental design is well structured and includes a clear methodology for evaluating different machine learning models. The use of standard metrics (e.g. accuracy, precision, recall) to evaluate model performance ensures that the evaluation is comparable to other studies. The inclusion of a self- supervised strategy to deal with missing data demonstrates an advanced approach to improving model accuracy.

Reproducibility:
Making the dataset and code available for public access will improve the reproducibility of the research. By allowing other researchers to replicate and build on the work, authors ensure that their claims can be independently verified, further supporting the integrity of the submission.

Ethical considerations and potential biases:
While the technical aspects of correctness are well addressed, the paper could benefit from a more detailed exploration of potential biases in the data and the ethical implications of using machine learning in educational settings. Ensuring that the models do not reinforce existing inequalities or introduce new biases is necessary to enable responsible application of this research.

**Documentation:**

The documentation for the dataset appears to be thorough, including details on data collection, processing, and the handling of missing data. The authors also plan to make the data and code publicly available, which will support reproducibility and further research.

**Ethics:**

There are several ethical concerns that warrant further discussion:
-	Privacy and consent: ensuring that data collection complies with privacy regulations and that consent has been obtained from all participants or their legal guardians.
-	Bias and fairness: addressing potential bias in the data and ensuring that the models do not have a significant impact on certain groups of students.
-	Responsible use: Providing guidelines for responsible use of the dataset and models, particularly in education.
-	Impact of incorrect predictions: Considering the potential impact of incorrect predictions on students' educational progression and well-being.

**Limitations:**

The authors have partially addressed the limitations and potential negative societal impacts of their work. They mention the challenges of data collection and the inherent diversity in students with learning difficulties. However, a more detailed discussion of how to mitigate these limitations and the potential biases in the model predictions would strengthen the work.

**Opportunities For Improvement:**

One area for improvement is the discussion of the ethical and social implications of using machine learning in education. While the technical aspects are addressed, the potential risks and limitations, such as biases in the data and the impact of incorrect predictions, should be addressed more thoroughly.

**Relation To Prior Work:**

The paper clearly discusses how this work differs from previous contributions. It highlights the lack of publicly available datasets for predicting early reading achievement and positions the ECRI dataset as a significant advance in this area. The comparison with related work on machine learning in education and how to deal with missing data is well presented.

**Summary And Contributions:**

This paper presents the Enhanced Core Reading Instruction (ECRI) dataset, a large-scale longitudinal dataset collected from 6,916 first graders and 172 teachers at 44 schools. The aim of the study is to predict early childhood reading performance using state-of-the-art machine learning models. The main contributions include:
- Introduction of a novel dataset for predicting early reading achievement.
- Empirical evaluation of machine learning models for early childhood educational patterns.
- Demonstration of a self-supervised strategy using a multilayer perceptual network (MLP) to handle missing data and improve prediction accuracy.
- Public availability of the dataset and code to encourage further research and development in this area.

---

> ### Author Rebuttal · Authors · 2024-08-18
>
> Thank you for the thoughtful feedback and positive remarks regarding the originality, significance, and methodology of our work. Below, we address questions concerning bias, consent, and future plans. We note that our dataset, code, and models will be made publicly available to ensure reproducibility.
>
>
> # Q1: Biases and Ethical Implications
>
> **Q1.1: Potential Bias Mitigation:**
>
> We acknowledge the critical ethical and societal implications of using machine learning in education and the importance of discussing bias mitigation to minimize risks. We aim for machine learning predictions to serve as tools to assist teachers in identifying at-risk students for timely interventions. However, the benefits of such technology may not be realized, e.g., if the underlying dataset is biased. Even with diverse and representative datasets, model failure and misuse can potentially harm students who may already be at risk, such as through labeling or misidentification. We have expanded our discussion of data bias and model fairness in our response to [Reviewer hdhn] (Q2). We hope that the dataset can be used as a step toward better understanding and addressing bias issues at the intersection of ML and education.
>
>
>
> **Q1.2: Privacy and Consent:**
> In this study, parental consent was first obtained for all participating students. The students were then randomized and assigned unique identification numbers (IDs). These IDs were used to test students in schools, ensuring that all data collected remained confidential. The name-ID information was securely stored in a password-protected database, which was accessible only to team members listed on the Institutional Review Board (IRB) protocol, ensuring compliance with ethical guidelines for research. We are committed to regularly reviewing and improving our data collection methods to align with best practices.
>
>
>
> **Q1.3: Future Plans:**
> Thank you for asking! We plan to maintain and update the dataset in the future to include more diverse data. We have been constructing a similar-scale benchmark focused on mathematics, which we plan to release and use to develop ML-based approaches in a broader context. Additionally, we will update the introduced dataset to ensure its continued value to the scientific community. We are currently working on expanding the literacy dataset by including more schools and districts with diverse demographics. We aim to use these datasets to gain deeper insights into children’s learning development and responsiveness to interventions.
>
>
> **Q1.4: Potential Long-Term Impact:**
> We greatly appreciate the feedback. Early and accurate diagnosis of reading difficulties and predictions of whether a child would benefit from a particular intervention is crucial for preventing the negative consequences of a reactive "wait-to-fail" approach [1, 2]. Without early identification, children's struggles can be misinterpreted, leading to lowered self-esteem, feelings of inadequacy, and a higher likelihood of academic failure. Research has consistently shown that interventions are most effective when tailored to children's specific learning needs and implemented during kindergarten or first grade, a critical period of brain plasticity. Our introduced models can be used to predict student responsiveness, i.e., to an intervention, which can help teachers identify individualized support strategies. However, despite the benefits of integrating ML-based tools in real-world classrooms, it is critical to consider potential long-term impacts. For instance, labeling a child with a diagnosis carries risks, such as lowered expectations and the potential for the child to be defined by their difficulties. Alongside the release of our data and models, we will have users sign an agreement that includes a discussion on ethical use, known biases, and model failure analysis. Nonetheless, we believe that these risks can be effectively mitigated in an educational context through supportive practices and that the potential benefits of early and accurate identification—such as improved intervention outcomes, reduced long-term negative effects, and enhanced overall well-being—far outweigh these risks when managed appropriately.
>
> Thank you again for the feedback, which has improved the quality of our manuscript. We are happy to address any further questions or concerns.
>
> ------------------------------------------
> References for Reviewer pkea
>
> [1] S. Otaiba et al. ‘’Waiting to fail’ redux: Understanding inadequate response to intervention.’ Learning Disability Quarterly, 2014.
>
> [2] C. Reynolds and S. E. Shaywitz. ‘Response to Intervention: Ready or not? Or, from wait-to-fail to watch-them-fail.’ School Psychology Quarterly, 2009.

---

> > ### Author Response · Authors · 2024-08-30
> >
> > Dear Reviewer pkea,
> >
> > Thank you again for the valuable feedback and insights. We hope our rebuttal addresses your concerns. To ensure clarity on the point for bias mitigation (discussed in our response to Reviewer hdhn’s Q2), we incorporated ablations in the supplementary to investigate the role of various data sampling and loss balancing strategies, e.g.,
> >
> > | **Method**       |     | **Word Identification** |            |             |            | **Word Attack** |            |             |            |
> > | ---------------- | --- | ----------------------- | ---------- | ----------- | ---------- | --------------- | ---------- | ----------- | ---------- |
> > |                  |     | **High FRL**            |            | **Low FRL** |            | **High FRL**    |            | **Low FRL** |            |
> > |                  |     | Accuracy                | AUC        | Accuracy    | AUC        | Accuracy        | AUC        | Accuracy    | AUC        |
> > | Baseline     |     | **0.6217**              | 0.5717     | 0.6736      | 0.6417     | 0.6656          | 0.6178     | 0.5575      | 0.5298     |
> > | SMOTE [3]           |     | 0.6104                  | **0.6080** | 0.7000      | 0.6868     | 0.6558          | 0.6577     | **0.7273**  | **0.7195** |
> > | RandomOverSample  |     | 0.5844                  | 0.5835     | **0.7273**  | **0.7181** | **0.6818**      | **0.6832** | 0.6545      | 0.6495     |
> > | CB-Loss [4]         |     | 0.5909                  | 0.5845     | 0.7091      | 0.7090     | 0.6753          | 0.6770     | 0.6455      | 0.6369     |
> >
> >
> > Please let us know if you still have outstanding concerns. We will be happy to address them.
> >
> >
> > [3] A. Fernández et al. ‘SMOTE for learning from imbalanced data: progress and challenges, marking the 15-year anniversary.’ Journal of Artificial Intelligence Research, 2018.
> >
> > [4] Y. Cui, et al., Class-Balanced Loss Based on Effective Number of Samples, CVPR 2019.

---

### Official Review · Reviewer_4e3s · 2024-07-19
**An important dataset for predicting student reading performance**

**Rating:** 7
**Confidence:** 4
**Correctness:** Claims are supported in the paper.
**Clarity:** The paper is very easy to follow and …

**Review:**

While the paper’s main contribution is the ECRI dataset, it would be nice to further strengthen the performance advantage of MaskMLP by conducting statistical tests whether the improvement is significantly higher than the other model’ performance or not. This may not be the highlighted contribution but it would definitely be adopted by future works if it’s better than VIME or SCARF due to its simplicity.

The paper can benefit from a well-designed discussion and descriptive visualizations of the diverse demographic variables and reading performance predictors from the dataset. As of now, the only reported visualization is from the appendix which is an aggregated one (Fig 1 in Appendix). The goal for this track is to sell/showcase how important your dataset is as a contribution and what it encompasses which can be adopted by future works. For example, can you show visualizations of how ethinicity/gender in related to Stanford Achievement Tests or Woodcock Master Tests? Moreover, how does the best model of MaskMLP perform with these diverse groups? I believe including these forms of analysis will provide more insights to the readers.

I’m not quite confident with using one method of feature importance from a model. A common challenge with feature importance algorithms is that they yield different ranks/combinations. The authors can explore comparing the results from Figure 4 with feature importance methods from other algorithms such as Random Forest (or any from the previous methods VIME/SCARF) to compare if some overlap of the ranking is preserved or not and discuss how this might be.

Aside from the points raised above, I do not have any major concerns towards rejection of the paper.

**Strengths:**

The paper is very much readable and has the level of clarity typically understood by a majority of the ML community. I do agree that large scale diverse dataset is extremely scarce particularly datasets that can be used to directly train models. The MaskMLP method seems to be a simple yet effective for the challenge of missing features as it seems to perform better than traditional ML models and previous works’ methods. Structure-wise, the paper can achieve the level of completeness required for NeurIPS upon completion of the suggestions in this review + other reviewers.

**Additional Feedback:**

See main review.

**Documentation:**

The dataset should be hosted in an open repository.

**Ethics:**

No issues.

**Limitations:**

The limitation section seems aptly discussed.

**Opportunities For Improvement:**

As summarized from above:

1. The dataset should be converted into different machine readable formats such as Croissant format or JSON. It should also be hosted in an open repository.
2. The discussion about MaskMLP needs to be expanded. Indicate if this is a novel method as a solution for missing data and supporting statistical tests comparing performances of other methods.
3. Improved expanded discussions of relationships of diverse demograph variables and reading performance variables as a way to present to readers what the data has.
4. Improved feature importance analysis by comparing results to off-the-shelf methods such as from Random Forest.

**Relation To Prior Work:**

No issues with prior work.

**Summary And Contributions:**

The paper introduce the Enhanced Core Reading Instruction (ECRI) dataset which is a large-scale longitudinal tabular dataset collected across 44 schools with 6,916 students and 172 teachers aimed for the task of student reading performance prediction. The authors motivate the need for the dataset in line with the importance of automatically assessing a learner’s skill in reading as well as the limited research on exploring how ML models can be used for large-scale education datasets with diverse student profiles. The paper also introduces a new pretraining approach for MLP (called MaskMLP) which aims to alleviate the missing data problem commonly observed with large educational datasets.

---

> ### Author Rebuttal · Authors · 2024-08-18
>
> Thank you for your thoughtful feedback; we have incorporated the requested discussions and clarifications, as outlined below.
>
> # Q1: Statistical Tests
>
> We have revised Sec. 4 and Sec. 5 to clarify the significance of the results and method. Specifically, we have added statistical testing using a paired samples t-test [8, 9] in Sec. 5. We compare MaskMLP against MLP and VIME over five different folds to determine if MaskMLP significantly outperformed the other models. For the MLP baseline, the t-statistic was 3.876, and the p-value was 0.0090, suggesting that MaskMLP's improvements are statistically significant. For VIME [1], the t-statistic was 2.1023, and the p-value was 0.0517, indicating a trend toward significance. When comparing with SCARF [2], the approaches differ in their loss functions (as further discussed in Q4.2). The t-statistic in word identification was 2.0954, and the p-value was 0.0521.
>
> # Q2: Demographic Variables and Visualizations
>
> Thank you for the comments. As suggested by the reviewer, we have added visualizations illustrating the relationship between gender and reading performance predictors from the dataset (see the attached document). We have also included a discussion regarding gender, at-risk students, and low socioeconomic school-based breakdowns; please refer to our response to [Reviewer hdhn] (Q2). We agree that added visualizations improve clarity and better showcase our dataset contribution.
>
> # Q3: Feature Importance Ablation
>
> To analyze the extent of model-dependence in feature importance analysis, as suggested, we have incorporated feature importance results based on XGBoost in the revised supplementary (shown in the attached rebuttal document, hyperparameters: learning_rate: 0.15, num_leaves: 36, max_depth: 6, max_bins: 100). We do find some consistency with the predictive analysis in Fig. 4 of the main paper. For instance, word identification at the onset of the school year remains the most influential, as does word attack assessment. Moreover, gender remains the lowest selected feature. Based on XGBoost’s feature importance analysis, we find that in the intervention group, student baseline reading skills were the most influential features. In contrast, for the control group, baseline reading skills are still important classroom features, such as teacher knowledge, classroom management behaviors, and fidelity of curriculum implementation, which also played significant roles in differentiating student responses. We now discuss this in our supplementary.
>
>
> # Q4: Data and Method Clarifications
>
> **Q4.1: Croissant Documentation:**
> Our original submission contained a zip with a Croissant document, which can also be accessed at the following link:
>
> https://github.com/ECRIanonymous/ECRI-Dataset
>
> We are currently working on adding data loading, training, and evaluation code.
>
> **Q4.2: MaskMLP Clarifications:**
>
> We have clarified that MaskMLP studies self-supervised pre-training for handling missing data, which has not been analyzed in related work, such as SCARF [2]. MaskMLP operates by reducing the distance between missing and non-missing data in the embedding space, i.e., in contrast to many current approaches that focus on input interpolation [3-6].
>
> Nonetheless, MaskMLP and SCARF [2] can both be considered variants of traditional self-supervised learning (e.g., BERT [10], which masks words for pre-training, similarly to our vector masking strategy). However, as mentioned, BERT and SCRAF have not been studied in educational contexts nor as mechanisms for dealing with missing data. Moreover, SCRAF leverages contrastive and triplet loss terms in pre-training, whereas we incorporate cosine-based loss, which has been shown to be more effective (Table 2 in the supplementary material). The loss function change outperforms SCARF (e.g., by 8.7% in word identification skill prediction accuracy for the intervention group).  This finding also suggests that our benchmark introduces orthogonal challenges to ones in tabular datasets from prior work, such as the ones used to validate SCARF in [2]. We have clarified this in the paper.
>
>
> Thank you again for the helpful suggestions, which have improved the clarity and impact of our work.
>
>
>  -----------------------------------------------
>
>  References for Reviewer 4e3s
>
> [1] J. Yoon et al. ‘VIME: Extending the success of self-and semi-supervised learning to tabular domain.’ Advances in Neural Information Processing Systems, 2020.
>
> [2] Bahri Dara et al. ‘SCARF: Self-supervised contrastive learning using random feature corruption.’ International Conference on Learning Representations, 2022.
>
> [3] J. Yoon et al. 'Gain: Missing data imputation using generative adversarial nets.’ International Conference on Machine Learning, 2018.
>
> [4] J. You, et al. 'Handling missing data with graph representation learning.' Conference on Neural Information Processing Systems, 2020.
>
> [5] S. Zheng et al. 'Diffusion models for missing value imputation in tabular data.' arXiv, 2022.
>
> [6] R. D. Camino et al. 'Improving missing data imputation with deep generative models.' arXiv, 2019.
>
> [7] Markelle Kelly et al. 'The UCI Machine Learning Repository' https://archive.ics.uci.edu
>
> [8] T. G. Dietterich. ‘Approximate statistical tests for comparing supervised classification learning algorithms.’ Neural Computation, 1998.
>
> [9] J. Demsar. ‘Statistical comparisons of classifiers over multiple data sets.’ Journal of Machine Learning Research, 2006.
>
> [10] J. Devlin et al. ‘BERT: Pre-training of deep bidirectional transformers for language understanding.’ Association for Computational Linguistics, 2019.
>
> [11] L Breiman. ‘Statistical modeling: The two cultures.’ Quality Control and Applied Statistics, 2003.

---

> > ### Comment · Reviewer_4e3s · 2024-08-18
> > **Acknowledgment of response**
> >
> > Dear authors,
> >
> > Thank you for the response to my review. This is to acknowledge that I have read said response as well as the responses to my fellow reviewer's questions/feedback. I trust the authors will reflect my suggested additions on certain parts of the paper including the MaskMLP method and feature importance and its consistency. I will keep my score as it's already in favor of acceptance (7).

---

### Official Review · Reviewer_hdhn · 2024-07-25

**Rating:** 7
**Confidence:** 2
**Correctness:** Yes
**Clarity:** Yes

**Review:**

1. The methodology is well-documented, including rigorous data collection and preprocessing steps.
2. The paper is well-organized, with clear sections on the dataset, methodology, and experimental results.
3. The paper could better address potential biases in the dataset and their implications for model performance and fairness.
4. The paper could better offer step-by-step examples or tutorials for using the dataset and models.

**Strengths:**

1. The submission makes a highly significant contribution by introducing the Enhanced Core Reading Instruction (ECRI) dataset. This dataset is one of the first large-scale, longitudinal datasets focused on early childhood reading performance. By providing comprehensive demographic, assessment, and intervention data, the dataset enables the development and evaluation of machine learning models for predicting reading performance.

2. The research quality is evident in the thorough documentation of the data collection process, the rigorous evaluation of multiple machine learning models, and the innovative approach to handling missing data. The authors demonstrate a comprehensive understanding of the challenges associated with early childhood reading assessment and provide a robust solution in the form of the MaskMLP model. The experimental results are well-presented, showing the superior performance of the proposed model over several strong baselines.

**Additional Feedback:**

No

**Documentation:**

Yes

**Limitations:**

The authors addressed the limitations.

**Opportunities For Improvement:**

1. While the introduction of the ECRI dataset is a significant contribution, its impact could be limited by the scope of the dataset. The dataset focuses on early childhood reading performance, which, although important, may not address the needs of other critical areas in education, such as mathematics or social-emotional learning. Expanding the dataset to include a broader range of subjects and skills would enhance its overall significance and applicability.

2. More detailed analysis of how the models perform across different demographic groups (e.g., by race, gender, socio-economic status) would provide a clearer understanding of potential biases. The paper could discuss strategies to mitigate these biases and ensure fairness.

3. The dataset may not fully capture the diversity of student experiences and educational settings. Ensuring that the dataset and models are representative of diverse populations is crucial for equitable outcomes.

**Relation To Prior Work:**

Yes

**Summary And Contributions:**

A comprehensive longitudinal dataset focusing on early childhood reading performance. It collected from 44 schools, encompassing 6,916 students and 172 teachers and includes various demographic, assessment, and intervention data.

Proposes a self-supervised pre-training technique (MaskMLP) to manage missing data without explicit imputation. Shows improved performance in handling partial and diverse data compared to other models.

---

> ### Author Rebuttal · Authors · 2024-08-18
>
> Thank you for the constructive feedback and remarks on the significance of the contribution, research quality, methodology, and paper organization. Below, we address concerns regarding the scope and additional analysis.
>
> #  Q1: Focus on Literacy vs. Other Areas
>
> We see the need for diverse early childhood education datasets that can support data-driven insights and models. However, while our work provides a focused and in-depth foundation within one area of learning and development, pursuing full coverage is difficult as large-scale longitudinal studies with young children involve an inherently complex data collection process.
>
> As suggested by the reviewer, we have revised Sec. 1 and Sec. 2 to highlight further that (1) there is a general lack of publicly available machine learning-suitable benchmarks&mdash;standardized open data sharing practices have not yet been adopted in the field of education, and (2) our focus on literacy was motivated by its critical role in shaping life outcomes, including academic success, vocational opportunities, economic stability, and social well-being [1]. The latter was a pragmatic design choice, as literacy challenges can lead to significant social-emotional and mental health issues, such as increased risks of anxiety and depression [2-4], and strongly influence educational attainment (a key predictor of overall health and longevity [5,6]). Yet, 65% of fourth graders not reading at grade level and struggling readers are more likely to drop out of school, particularly among African-American and Hispanic students [2]. Such statistics motivated us to pursue a focused, scalable study into this impactful domain.
>
> We agree that expanding our dataset is a natural next step. We are indeed working to apply our methodologies to similar-scale datasets related to mathematical learning (however, this effort is still ongoing). Based on the suggestion, we have expanded the discussion in Future Work (Sec. 7) to note that replication of our modeling approach across other datasets is critical to establishing the generalizability of the findings, including additional educational areas, various developmental backgrounds and student identities (including ethnicity, gender, race, disability, and their intersections), and different interventional contexts.
>
> # Q2: Group Analysis and Bias Characterization
>
> Thank you for the suggestion. To improve bias characterization and mitigation in our context, we have incorporated a breakdown of three specific types of demographic groups, discussed below.
>
> **Q2.1: Gender-based Breakdown:** We have added a discussion in the supplementary based on a gender-based breakdown of the results:
>
> | **Method**   |   | **Word Identification** |   |  |   |  **Word Attack** |   |   |   |
> |--------------|---|------------------------ |---|--|---|------------------|---|---|---|
> |              |   | **All Students**        |   | **Intervention Group** | | **All Students** |   | **Intervention Group** |
> |              |   | Accuracy  | AUC         | Accuracy | AUC          | Accuracy | AUC        | Accuracy | AUC      |
> | **School Split** |   |   |   |   |   |   |   |   |   |
> | Overall      |   | 0.6726    | 0.6693      | 0.7704   | 0.7633       | 0.6697   | 0.6545     | 0.7704   | 0.6869   |
> | Male         |   | 0.6753    | 0.6753      | 0.7784   | 0.7698       | 0.6736   | 0.6572     | 0.7585   | 0.6896   |
> | Female       |   | 0.6636    | 0.6606      | 0.7620   | 0.7521       | 0.6422   | 0.6486     | 0.7765   | 0.6754   |
> | **Student Split** |   |   |   |   |   |   |   |   |   |
> | Overall      |   | 0.6919    | 0.6899      | 0.7766   | 0.7537       | 0.6741   | 0.6485     | 0.7865   | 0.7333   |
> | Male         |   | 0.6969    | 0.6968      | 0.7864   | 0.7666       | 0.6782   | 0.6496     | 0.7725   | 0.7202   |
> | Female       |   | 0.6728    | 0.6755      | 0.7620   | 0.7425       | 0.6706   | 0.6349     | 0.7886   | 0.7426   |
>
> Interestingly, despite the fairly balanced distribution (L. 164 in the paper, 51.32% male and 48.68% female), we observe differing performances among the groups. However, inspecting the overall statistics between the two groups in the dataset (shown in the attached rebuttal PDF, now incorporated into the supplementary material) shows generally comparable performance. Specifically, in word identification, we observe the model underperforming in females. The reasons for this gap in model performance are likely complex, potentially involving various personal and environmental factors, yet current literature is insufficient to fully explain this finding. We have revised the discussion in our paper to mention that there are known gender differences in the characteristics and identification of learning disabilities; females are more likely to compensate, i.e., use alternative learning strategies, than males, and are under-referred to diagnostic categories [17] (e.g., due to symptom internalization, thus receiving less attention in instructional contexts and potentially contributing to the model’s reduced accuracy). Based on prior research, females likely use different strategies than males to achieve task performance, especially in the intervention group. Some studies also suggest females may slightly outperform males and make greater literacy gains during early grades [18-21]. The model may misclassify females at a higher rate by overemphasizing the role of initial low scores in females (as shown in Fig. 4 of the main paper). The previously identified use of compensatory or alternative strategies for reading task performance in females can further confound the model; however, this requires further study in the future. We have revised Sec. 5 and the supplementary material to incorporate additional discussion on this observed bias.
>
> In addition to gender-based breakdown, we incorporate two additional groups: at-risk students and socio-economic status, defined in Q 2.2 and Q2.3.

---

> > ### Author Rebuttal · Authors · 2024-08-18
> >
> > **Q2.2: At-Risk Students:** We have added results over the at-risk student population (SAT below the 30th percentile, out of which 7.4% of the students received special education services and 19.7% were English Learners):
> >
> > | **Method**   |   | **Word Identification** |   |  |   |  **Word Attack** |   |   |   |
> > |--------------|---|------------------------ |---|--|---|------------------|---|---|---|
> > |              |   | **All Students**        |   | **Intervention Group** | | **All Students** |   | **Intervention Group** |
> > |              |   | Accuracy  | AUC         | Accuracy | AUC          | Accuracy | AUC        | Accuracy | AUC      |
> > | **School Split** |   |   |   |   |   |   |   |   |   |
> > | Overall      |   | 0.6726    | 0.6693      | 0.7704   | 0.7633       | 0.6697   | 0.6545     | 0.7704   | 0.6869   |
> > | At-Risk      |   | 0.6692    | 0.6672      | 0.7195   | 0.7140       | 0.6436   | 0.6170     | 0.7326   | 0.6429   |
> > | **Student Split** |   |   |   |   |   |   |   |   |   |
> > | Overall      |   | 0.6919    | 0.6899      | 0.7766   | 0.7537       | 0.6741   | 0.6485     | 0.7865   | 0.7333   |
> > | At-Risk      |   | 0.6691    | 0.6693      | 0.7096   | 0.7043       | 0.6707   | 0.6434     | 0.7558   | 0.6731   |
> >
> > We found a concerning pattern of lower model performance over at-risk students. The results are more pronounced than those of the gender-based analysis, which is expected due to the heterogeneous population of at-risk students, which can confound ML models. At-risk students often face multiple types of challenges that influence their learning trajectories, such as language barriers, learning disabilities, and socio-emotional issues (further discussed in Q2.3). These multifaceted complexities make it difficult for an ML model to predict accurate outcomes, thereby failing to effectively support a student population that could benefit the most from such technologies.
> > Given the critical role that ML models can play in educational interventions, it's essential to improve their performance for at-risk students. This could involve incorporating more comprehensive data, considering broader contextual factors, and designing models that can robustly account for the diverse challenges faced by various student groups.
> >
> > **Q2.3: Socio-Economic Status (SES):** Regarding additional demographic groups, as part of the study, we have tried to obtain race-ethnicity and free/reduced-price lunch (FRL) status from participating schools and districts. However, we received inconsistent data as many schools did not have reliable information and were making assumptions, often based on unreliable sources (e.g., surnames). Additionally, some districts refused to share information due to confidentiality concerns. Thus, we decided only to use and release school-level race-ethnicity and FRL information.
> >
> > To analyze the role of SES in biasing model performance, we included an analysis based on the five schools with the highest FRL rate (see Fig. 1 in the supplementary, school IDs 12, 16, 15, 32, 41) and the five lowest (school IDs 3, 27, 30, 42, 44):
> >
> > | **Method**   |   | **Word Identification** |   |  |   |  **Word Attack** |   |   |   |
> > |--------------|---|------------------------ |---|--|---|------------------|---|---|---|
> > |              |   | **All Students**        |   | **Intervention Group** | | **All Students** |   | **Intervention Group** |
> > |              |   | Accuracy  | AUC         | Accuracy | AUC          | Accuracy | AUC        | Accuracy | AUC      |
> > | **School Split** |   |   |   |   |   |   |   |   |   |
> > | High FRL |   | 0.6200    | 0.5770      | 0.7500   | 0.7624       | 0.6281   | 0.6178     | 0.6042   | 0.5556   |
> > | Low FRL | | 0.6874    | 0.6979      | 0.5873   | 0.6129       | 0.5574   | 0.5297     | 0.6785   | 0.6066   |
> > | **Student Split** |   |   |   |   |   |   |   |   |   |
> > | High FRL  |   | 0.6217    | 0.5717      | 0.7500   | 0.7579       | 0.6656   | 0.6178     | 0.6301   | 0.6048   |
> > | Low FRL | | 0.6736    | 0.6417      | 0.7143   | 0.6667       | 0.5575   | 0.5298     | 0.7358   | 0.6128   |
> >
> > where we observe a consistent trend of higher model prediction performance in schools with **higher SES** (i.e., **less FRL**). We have included the distribution of student race/ethnicities within these two groups in an attached pdf, as well as in our supplementary revision. The two groups contain a significantly different distribution of SES and race/ethnicity factors (e.g., Hispanic student population of 8% vs. 35% and African American of 2% vs. 7%). This is an interesting finding, as higher SES students may have access to other resources beyond in-school instruction, e.g., private tutoring. However, we find much higher predictability within high SES, despite the potential presence of such external factors.
> >
> > We have incorporated a discussion on this concerning trend in our supplementary, as well as how factors related to retention and fewer external challenges may all play a role in contributing to the lower model performance. With the introduced benchmark and uncovered challenges, we hope to engage ML researchers, practitioners, and educators in educational AI bias and fairness issues, facilitating more precise modeling and responsible use.
> >
> > We further observe that teacher knowledge and experience variability had a markedly reduced impact on student outcomes in classrooms that received the intervention. This contrasts with classrooms that did not receive the intervention and instead used a range of instructional approaches commonly found across U.S. schools, where such variability has a more pronounced effect. These findings suggest that consistently applying evidence-based strategies can help mitigate differences in educational delivery, leading to more equitable outcomes. In Sec. 2, we now also discuss how inherent dataset imbalance and task diversity may cause ML models to learn and amplify harmful biases, i.e., result in disparate outcomes between students and groups.

---

> > > ### Author Rebuttal · Authors · 2024-08-18
> > >
> > > **Q2.4: Bias Mitigation:** As our large-scale dataset comprises learners with diverse skills and backgrounds, it can be used to `provide a clearer understanding of potential biases’ (as the reviewer notes) and analyze mitigation techniques in educational contexts. We have added a discussion on the need to address the aforementioned predictive fairness gaps through effective bias mitigation strategies in future work and also note that addressing bias and predictive fairness in ML models often involves careful dataset handling [8, 9, 10] and model optimization [11, 12] or calibration [13] strategies. We have experimented with standard approaches for bias mitigation, such as the Synthetic Minority Oversampling Technique (SMOTE) [8]. However, we did not observe a consistent improvement over the groups studied in our data. We will clarify this in the paper. Moreover, while MaskMLP can be used to avoid sparse datasets biasing the model, balancing fairness and model accuracy can be complex in practice.
> > >
> > > # Q3: Representivness
> > > We have expanded the discussion regarding educational settings **(Q1, above)** and representativeness **(Q2, above)** (the large-scale dataset reflects US census for certain categories, e.g., Asian and Hispanic, but does under-represent others, e.g., African American).
> > >
> > >
> > > We thank the reviewer for the suggestions, which have improved the quality of the work. We hope this clarifies the concerns raised and welcome further suggestions and comments to strengthen our paper.
> > >
> > >
> > >
> > >
> > > ------------
> > >  References for Reviewer hdhn
> > >
> > >
> > > [1] L. Irwin et al. ‘Early child development: A powerful equalizer.’ Human Early Learning Partnership, 2007.
> > >
> > > [2] D. J. Hernandez. ‘Double jeopardy: How third-grade reading skills and poverty influence high school graduation.’ Annie E. Casey Foundation, 2011.
> > >
> > > [3] R. L. Hendren et al. ‘Recognizing psychiatric comorbidity with reading disorders.’ Frontiers in Psychiatry, 2018.
> > >
> > > [4] D. Mugnaini et al. ‘Internalizing correlates of dyslexia.’ World Journal of Pediatrics, 2009.
> > >
> > > [5] L. Vernon-Feagans et al. ‘Targeted reading intervention: A coaching model to help classroom teachers with struggling readers.’ Learning Disability Quarterly, 2012.
> > >
> > > [6] V. Johnston. ‘Dyslexia: What reading teachers need to know.’ The Reading Teacher, 2019.
> > >
> > > [7] J. Rothwell. ‘Assessing the economic gains of eradicating illiteracy nationally and regionally in the United States.’ Barbara Bush Foundation for Family Literacy, 2020.
> > >
> > > [8] A. Fernández et al. ‘SMOTE for learning from imbalanced data: progress and challenges, marking the 15-year anniversary.’ Journal of Artificial Intelligence Research, 2018.
> > >
> > > [9] E. Ferrara.’"Fairness and bias in artificial intelligence: A brief survey of sources, impacts, and mitigation strategies.’ Sci, 2023.
> > >
> > > [10] S. Kotsiantis et al. ‘Handling imbalanced datasets: A review.’ GESTS International Transactions on Computer Science and Engineering, 2006.
> > >
> > > [11] T. Kamishima et al. ‘Fairness-aware classifier with prejudice remover regularizer.’ Machine Learning and Knowledge Discovery
> > > in Databases, 2011.
> > >
> > > [12] N. Grgić-Hlača et al. ‘Beyond distributive fairness in algorithmic decision making: Feature selection for procedurally fair learning.’  AAAI Conference on Artificial Intelligence, 2018.
> > >
> > > [13] M. Hardt et al. ‘Equality of opportunity in supervised learning.’ Advances in Neural Information Processing Systems, 2016.
> > >
> > > [14] O. Ozernov‐Palchik et al. ‘The relationship between socioeconomic status and white matter microstructure in pre‐reading children: A longitudinal investigation.‘ Human Brain Mapping, 2019.
> > >
> > > [15] N. Gaab, Nadine and Y. Petscher. ‘Screening for early literacy milestones and reading disabilities: The why, when, whom, how, and where.’ Perspectives on Language and Literacy, 2022.
> > >
> > > [16] Alison E. Baroody et al. 'Measures of preschool children's interest and engagement in literacy activities: Examining gender differences and construct dimensions.' Early Childhood Research Quarterly, 2013
> > >
> > > [17] D. L. Lefly and B. F. Pennington. ‘Spelling errors and reading fluency in compensated adult dyslexics.’ Annals of dyslexia, 1991.
> > >
> > > [18] D. Ready et al. ‘Explaining girls’ advantage in kindergarten literacy learning: Do classroom behaviors make a difference?.’ The Elementary School Journal, 2005.
> > >
> > > [19] J. Fonseca et al. ‘Girls have academic advantages and so do boys: A multicountry analysis of gender differences in early grade reading and mathematics outcomes.’ RTI Press, 2023.
> > >
> > > [20] E. McTigue et al. ‘Gender differences in early literacy: Boys’ response to formal instruction.’ Journal of Educational Psychology, 2021.
> > >
> > > [21] F. Borgonovi et al. ‘The evolution of gender gaps in numeracy and literacy between childhood and young adulthood.’ Economics of Education Review, 2021.

---

> > > > ### Comment · Reviewer_hdhn · 2024-08-29
> > > >
> > > > Thanks author for the detailed rebuttal, my concern was addressed and I will raise my score.

---

### Author Rebuttal · Authors · 2024-08-18

We thank the reviewers for their thoughtful feedback and remarks on the significance of our work.

The three reviewers note the contribution of the introduced dataset (e.g., ‘highly significant contribution’ [Reviewer hdhn], ‘important dataset’ [Reviewer 4e3s], ‘unique, large scale data set’ providing a ‘solid foundation for predictive modeling’ [Reviewer pkea]). Additionally, the reviewers highlight the clarity of the writing and the method (‘innovative approach’ [Reviewer hdhn], ‘simple yet effective’ [Reviewer 4e3s]). Reviewer hdhn and Reviewer pkea further note the ‘rigorous’ and ‘comprehensive’ evaluations.

We have incorporated the suggestions to improve the quality of our paper, summarized in individual responses below. Please also see our attached pdf with additional analysis and dataset statistics.

---

### Decision · Program_Chairs · 2024-09-26

**Decision:**

Accept (Poster)

**Comment:**

This paper presents a comprehensive longitudinal dataset (ECRI) focused on early childhood reading performance. The dataset, collected from 44 schools, includes data from 6,916 students and 172 teachers, along with demographic information, assessments, and intervention data. The authors propose a self-supervised pre-training technique to address the issue of missing data and employ several baselines in their experiments. While all reviewers agree that the paper is well-written and easy to follow, they have offered several suggestions to enhance the quality of the presentation. We encourage the authors to incorporate the rebuttal content into the final version.